# Simulation-based inference for efficient identification of generative models in computational connectomics

Jan Boelts[1,2]*, Philipp Harth[3], Richard Gao[1,2], Daniel Udvary[4], Felipe Yáñez[4], Daniel Baum[3], Hans-Christian Hege[3], Marcel Oberlaender[4,5], Jakob H. Macke[1,2,6]

**1** Machine Learning in Science, University of Tübingen, Tübingen, Germany, **2** Tübingen AI Center, University of Tübingen, Tübingen, Germany, **3** Department of Visual and Data-centric Computing, Zuse Institute Berlin, Berlin, Germany, **4** In Silico Brain Sciences, Max Planck Institute for Neurobiology of Behavior – caesar, Bonn, Germany, **5** Department of Integrative Neurophysiology, Center for Neurogenomics and Cognitive Research, Free University Amsterdam, Amsterdam, Netherlands, **6** Empirical Inference, Max Planck Institute for Intelligent Systems, Tübingen, Germany

* jan.boelts@mainbox.org

**Data Availability Statement:** Data and code for reproducing the results are available at https://github.com/mackelab/sbi-forconnectomics, including a tutorial on how to apply SBI in

## Abstract

Recent advances in connectomics research enable the acquisition of increasing amounts of data about the connectivity patterns of neurons. How can we use this wealth of data to efficiently derive and test hypotheses about the principles underlying these patterns? A common approach is to simulate neuronal networks using a hypothesized wiring rule in a generative model and to compare the resulting synthetic data with empirical data. However, most wiring rules have at least some free parameters, and identifying parameters that reproduce empirical data can be challenging as it often requires manual parameter tuning. Here, we propose to use simulation-based Bayesian inference (SBI) to address this challenge. Rather than optimizing a fixed wiring rule to fit the empirical data, SBI considers many parametrizations of a rule and performs Bayesian inference to identify the parameters that are compatible with the data. It uses simulated data from multiple candidate wiring rule parameters and relies on machine learning methods to estimate a probability distribution (the 'posterior distribution over parameters conditioned on the data') that characterizes all data-compatible parameters. We demonstrate how to apply SBI in computational connectomics by inferring the parameters of wiring rules in an *in silico* model of the rat barrel cortex, given *in vivo* connectivity measurements. SBI identifies a wide range of wiring rule parameters that reproduce the measurements. We show how access to the posterior distribution over all data-compatible parameters allows us to analyze their relationship, revealing biologically plausible parameter interactions and enabling experimentally testable predictions. We further show how SBI can be applied to wiring rules at different spatial scales to quantitatively rule out invalid wiring hypotheses. Our approach is applicable to a wide range of generative models used in connectomics, providing a quantitative and efficient way to constrain model parameters with empirical connectivity data.

computational connectomics in general. For running SBI using SNPE, posterior visualization, and posterior validation, we used the sbi package at https://github.com/mackelab/sbi. The benchmarking of SNPE and SMC-ABC methods, including the generation of reference posteriors, was performed using the sbibm package at https://github.com/sbi-benchmark.

**Funding:** This work was supported by the German Research Foundation (DFG; SPP 2041 PN 34721065; Germany's Excellence Strategy MLCoE-EXC number 2064/1 PN 390727645 to J.H.M.; SFB 1089 to M.O.), the German Federal Ministry of Education and Research (BMBF; project SiMaLeSAM, FKZ 01IS21055A and Tübingen AI Center, FKZ 01IS18039A to J.H.M.; grants BMBF/FKZ 01GQ1002 and 01IS18052 to M.O.), and the European Union's Horizon 2020 research and innovation program (grant agreement 633428 to M.O.; Marie Sklodowska-Curie grant agreement No. 101030918 to R.G.). The funders had no role in study design, data collection and analysis, decision to publish, or preparation of the manuscript.

**Competing interests:** The authors have declared that no competing interests exist.

## Author summary

The brain is composed of an intricately connected network of cells—what are the principles that contribute to constructing these patterns of connectivity, and how? To answer these questions, amassing connectivity data alone is not enough. We must also be able to efficiently develop and test our ideas about the underlying connectivity principles. For example, we could simulate a hypothetical wiring rule like "neurons near each other are more likely to form connections" in a computational model and generate corresponding synthetic data. If the synthetic, simulated data resembles the real, measured data, then we have some confidence that our hypotheses might be correct. However, the proposed wiring rules usually have unknown parameters that we need to "tune" such that simulated data matches the measurements. The central challenge thus lies in finding all the potential parametrizations of a wiring rule that can reproduce the measured data, as this process is often idiosyncratic and labor-intensive. To tackle this challenge, we introduce an approach combining computational modeling in connectomics, deep learning, and Bayesian statistical inference to automatically infer a probability distribution over the model parameters likely to explain the data. We demonstrate our approach by inferring the parameters of a wiring rule in a detailed model of the rat barrel cortex and find that the inferred distribution identifies multiple data-compatible model parameters, reveals biologically plausible parameter interactions, and allows us to make experimentally testable predictions.

## Introduction

Connectomics investigates the structural and functional composition of neural networks to distill principles of the connectivity patterns underlying brain function [1, 2]. Over the last years, advances in imaging and tracing techniques enabled the acquisition of increasingly detailed connectivity data [3–5] and led to significant insights [6–8]. These advances in data acquisition necessitate new computational tools for analyzing the data and testing hypotheses derived from it [9–11]. A recent computational approach for testing hypotheses in connectomics has been to use so-called *generative models* [12–14]. The idea of generative modeling is to develop a computational model capable of generating synthetic connectivity data according to a specific hypothesis, e.g., a wiring rule (Fig 1A, left). Subsequently, one can validate and refine the wiring rule (or the underlying computational model) by comparing the simulated with measured connectivity data (Fig 1A, right). Examples for this approach range from large-scale generative models of functional connectivity in the human cortex [15, 16], system-level network models of the mouse visual cortex [17], and generative models of cortical microcircuits [18].

As a specific example, we here consider a generative model for simulating hypothesized wiring rules in the rat barrel cortex [19]. The model is based on reconstructions of axon and dendrite morphologies from *in vivo* recordings [20] and reconstructions of the barrel cortex geometry, cytoarchitecture, and cellular organization [21, 22]. These anatomical features were combined into a 3D model to obtain a quantitative and realistic estimate of the dense neuropil structure for a large volume of the rat barrel cortex [19, 23]. Thus, by applying a hypothesized wiring rule to the structural features of the model, one can generate a corresponding synthetic barrel cortex connectome and compare it to empirical data to test the validity of the wiring rule. For example, [19] used the barrel cortex model to show that a wiring rule that only takes

**A** Conventional generative modeling

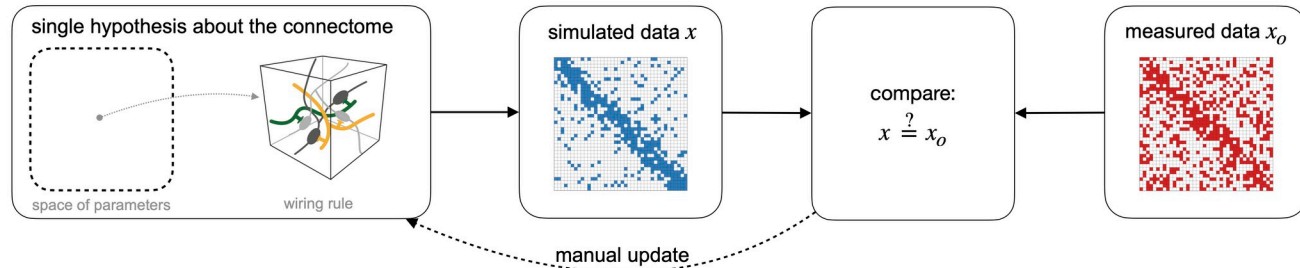

**B** Automated model identification with SBI

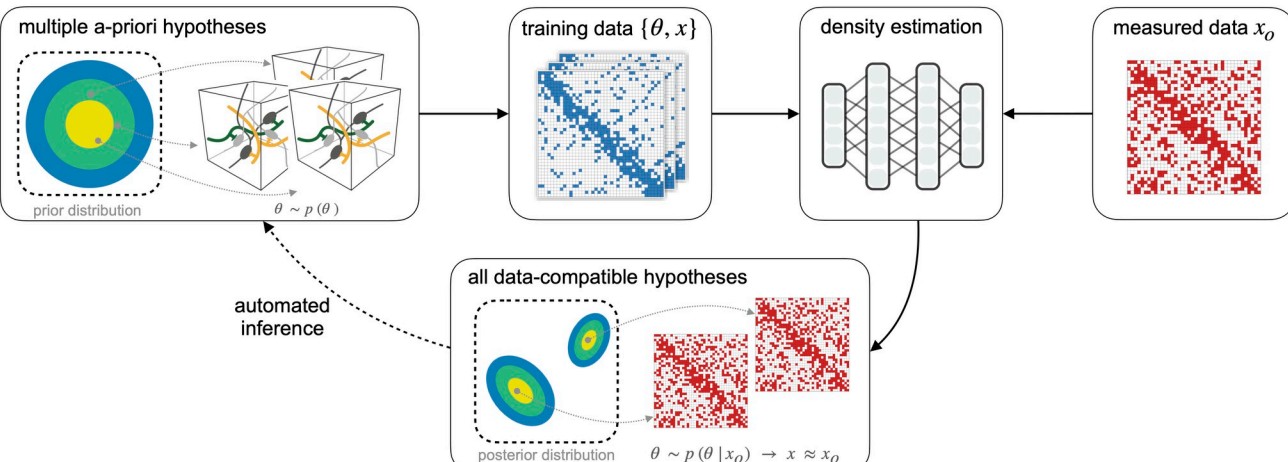

**Fig 1. Enhancing generative modeling in connectomics with simulation-based inference.** (**A**) Generative modeling is a common approach for testing hypotheses about the connectome: One implements a hypothesized wiring rule as a computational model that simulates connectivity data $x$ (left) and then tests and manually refines the rule by comparing simulated with measured data $x_o$ (right). (**B**) Our goal is to make this approach more efficient using simulation-based Bayesian inference (SBI): By equipping the generative model with parameters $\theta$, we define a space of multiple *a-priori* hypotheses (left) from which we can generate multiple simulated data $x$ (middle). We then use the simulated data to perform density estimation with artificial neural networks to estimate the *posterior distribution* over model parameters conditioned on the measured data, i.e., $p(\theta|x_o)$. The inferred posterior distribution characterizes *all* wiring rule parameters compatible with the measured data, replacing the manual refinement of single wiring rules in the conventional approach (bottom).

into account neuron morphology predicts connectivity patterns that are consistent with those observed empirically in the barrel cortex.

Building generative models that accurately reproduce connectivity measurements can be challenging: Suppose a hypothesized wiring rule does not reproduce the data. In that case, a common approach would be to introduce free parameters to the rule and employ a parameter-fitting algorithm to find the best-fitting wiring rule parameters (Fig 1A). However, identifying one specific wiring rule configuration for which simulated and empirical data match might not be sufficient: Given that the available empirical connectivity data is sparse compared to the structural and functional complexity of the connectome, it is likely that there are many data-compatible wiring rules. While some parameter fitting algorithms can be adapted to identify and compare multiple wiring rule configurations as well (see, e.g., [24]), they require a manual assessment of different solutions and generally involve many repetitions of the simulate-and-compare-to-measurements loop, which can be laborious and inefficient.

To address these challenges, we propose a new approach that employs Bayesian inference to enhance the identification of generative models in computational connectomics (Fig 1B). We achieve this by taking two conceptual steps: First, we equip the generative model with parameters $\theta$ and interpret different parameter combinations as variants of the underlying hypothesis, e.g., variants of the wiring rule. Second, we define a probability distribution $p(\theta)$ over the model parameters such that each parameter configuration corresponds to a different candidate wiring rule (Fig 1B, left), and use Bayesian inference to infer all data-compatible parameters. Given measured connectivity data $x_o$ and a parametrized generative model, we infer the conditional probability distribution over the model parameters given the measured data, i.e., the *posterior distribution $p(\theta|x_o)$*. The posterior distribution automatically characterizes *all* wiring rule parametrizations likely to explain the measured data. For example, by sampling different parameters from the inferred posterior we would obtain different wiring rule configurations all of which are likely to generate data similar to the measured data (Fig 1B, bottom). Additionally, the posterior distribution naturally quantifies uncertainties and correlations between the parameters, which can help to reveal parameter interactions and potential compensation mechanisms in the model.

On a technical level, standard Bayesian inference methods usually require access to the likelihood function of the model. However, generative models employed in computational connectomics are often defined as computer simulations for which the likelihood may not be easily accessible. Therefore, we propose using simulation-based inference (SBI, [25–29]). SBI enables Bayesian inference using only simulated data from the generative model, i.e., without requiring access to the likelihood. In particular, SBI performs conditional density estimation with artificial neural networks: It uses data simulated from the model to train an artificial neural network that takes data as input and predicts an approximation to the posterior distribution. Once trained on simulated data, the neural network can be applied to the measured data to obtain the desired posterior distribution $p(\theta|x_o)$ (Fig 1B, right).

We demonstrate our approach using the example of constraining wiring rules in the structural model of the rat barrel cortex introduced above. First, we show how to reformulate wiring rules as parametrized models to make them amenable to Bayesian inference. The resulting generative model consists of the parametrized wiring rule applied to the structural model to generate a simulated connectome of the rat barrel cortex. Second, we show that SBI can identify all parameter configurations that agree with measured connectivity data. When testing our approach in a scenario with simulated data and a known reference solution, we find that SBI performs accurately. In the realistic setting with measured connectivity data, SBI identifies a large set of rule configurations that reproduce observed and predict unobserved features of the connectome. Importantly, analyzing the inferred posterior reveals that this set of plausible rules is highly structured and reflects biologically interpretable interactions of the parameters. Finally, we illustrate the flexibility of the SBI approach by inferring two proximity-based wiring rules at different spatial scales to quantitatively show that Peters' rule cannot explain connectivity measurements in the barrel cortex.

Our approach provides a new quantitative and efficient tool for constraining model parameters with connectivity measurements and is applicable to many generative models used in connectomics. For example, it sets the stage for building generative models based on dense reconstructions of brain tissue [30–32] and inferring underlying connectivity principles using SBI. We are making all software tools required for applying SBI available via an established open-source software package for SBI (sbi, [33]), facilitating its use by researchers across the field.

## Results

### Formulating wiring rules in the rat barrel cortex as simulation-based models

To demonstrate the potential of simulation-based inference (SBI) for connectomics, we selected the problem of constraining wiring rules in the rat barrel cortex with empirical connectivity data. Applying SBI requires three ingredients: a simulation-based model with free parameters, a prior distribution over the parameters, and measured data (see Methods & materials for details). Our analyses are based on a digital model of the dense neuropil structure of the rat barrel cortex [19, 23], which we extended to obtain a simulation-based model. The model contains reconstructions of the number and distribution of somata, axon, and dendrite morphologies, and subcellular features like pre-synaptic boutons and post-synaptic dendritic spines. These anatomical features were collected for several neuron types in the barrel cortex and projecting neurons from the ventral posterior medial nucleus (VPM) of the thalamus and arranged in a 3D model (Fig 2A, see Methods & materials for details). The model enables us to construct a simulated connectome of the barrel cortex by applying a wiring rule that predicts the connectivity of each neuron pair in the model from the structural features [23]. Using this

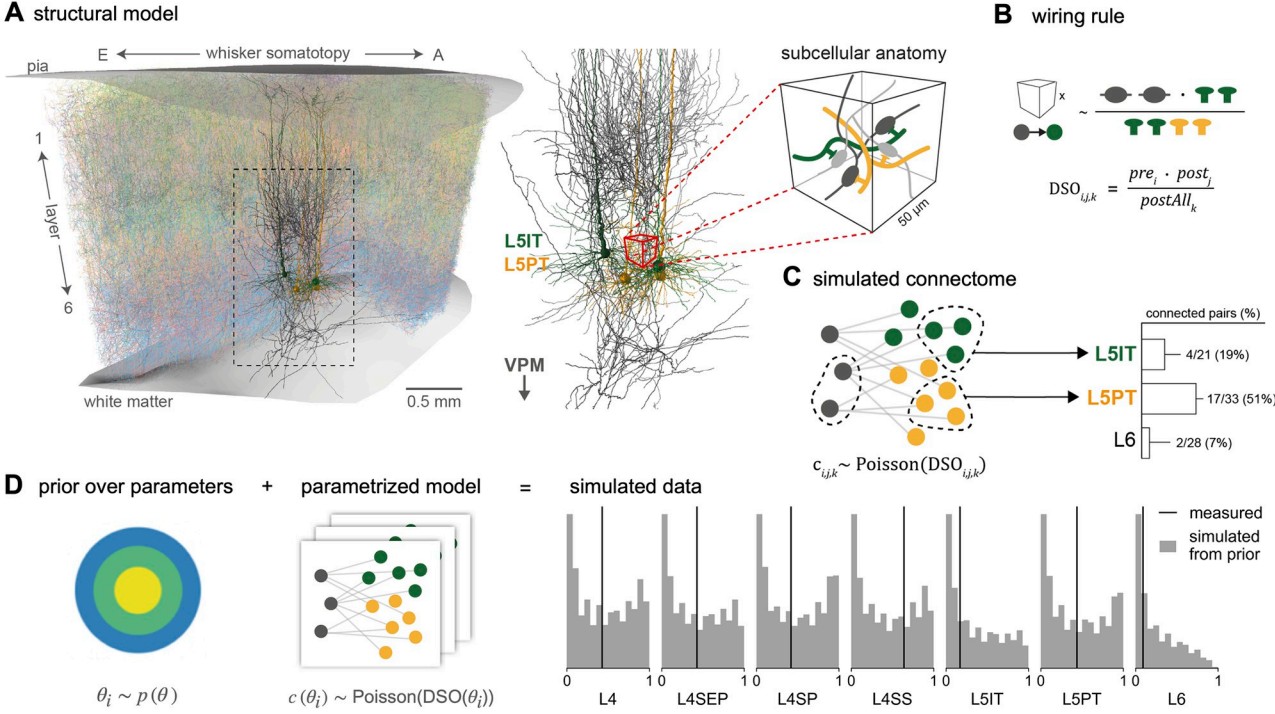

**Fig 2. Formulating wiring rules in the rat barrel cortex as simulation-based models.** (**A**) The structural model of the rat barrel cortex contains digital reconstructions of position, morphology, and subcellular features of several neuron types in the barrel cortex and the ventral posterior medial nucleus (VPM) of the thalamus. (**B**) We formulate a wiring rule that predicts the probability of a synapse between two neurons from their *dense structural overlap* (DSO), i.e., the product of the number of *pre-* and *post*synaptic structural features, normalized by all postsynaptic features in a given subvolume (*postAll*). (**C**) By applying the wiring rule to every neuron-pair subvolume combination of the model to connection probabilities and then sampling corresponding synapse counts from a Poisson distribution (left), we can simulate a barrel cortex connectome. To compare the simulated data to measurements, we calculate population connection probabilities between VPM and barrel cortex cell types as they have been measured experimentally (right). (**D**) To obtain a simulation-based model, we introduce parameters to the rule and define a prior distribution (left) such that each parameter combination corresponds to a different rule configuration and leads to different simulated connection probabilities (right, grey; measured data in black, [34, 35]).

approach, [19] proposed a parameter-free wiring rule that predicts barrel cortex connectivity from structural features like pre-synaptic boutons and postsynaptic dendritic spines. Here, we extended this wiring rule to a parameterized version that allows systematic analysis of how such structural features could interact and be predictive of connectivity.

**A wiring rule for the rat barrel cortex.** The parameter-free wiring rule introduced by [19] proposes that the probability of two neurons forming a synapse is proportional to a quantity called *dense structural overlap* (DSO). The DSO is defined as the product of the number of presynaptic boutons and postsynaptic contact sites (e.g., dendritic spines), normalized by the number of all postsynaptic targets in the neighborhood (denoted as $pre_i$, $post_j$ and $postAll_k$ respectively, for each neuron $i$, neuron $j$ and subvolume $k$ in the model, Fig 2B):

$$\text{DSO}_{i,j,k} = \frac{pre_i \cdot post_j}{postAll_k}. \tag{1}$$

To simulate a connectome from the DSO wiring rule, one applies the rule to every subvolume-neuron-pair combination of the structural model and then stochastically generates synapse counts by sampling from a Poisson distribution using the calculated synapse probabilities (Fig 2C, left, see Methods & materials for details). Ultimately, we want to evaluate the wiring rule against empirical data, i.e., compare the simulated connectome with measurements from the barrel cortex. To this end, the resulting simulated connectome can be used to calculate summary statistics in the format recorded in experiments, e.g., *in vitro* or *in vivo* paired recordings or dense reconstructions at electron microscopic levels (Fig 2C, right).

[19] showed that the DSO wiring rule can reproduce measured network characteristics at different scales. However, in its current form, the DSO rule assumes that the pre- and postsynaptic features have the same relative weight in determining the probability of a synapse and that these weights are the same across all barrel cortex cell types. Is this specific combination of pre- and postsynaptic features in the DSO rule the only valid choice? Specifically, it has been found that single boutons from VPM axons can establish multiple synapses in cortical layer four [36]; it has also been observed that in some cases, multiple synapses are formed at a single postsynaptic spine [37]. Thus, the weighting of the DSO features could, in principle, be unequal. A common approach to testing this question would be to iteratively modify the rule, e.g., by adding a scaling factor to the postsynaptic features or introducing different scaling factors for every cell type. However, this approach can be inefficient because any changes to the rule would require rerunning the procedure of generating simulated data and manually comparing it to measured data.

**Defining a wiring rule simulator.** In order to test different variations of the DSO rule efficiently, we introduced three parameters to the DSO rule and applied SBI to constrain these parameters with measured data. The three DSO parameters represent the relative weight with which each local subcellular feature contributes to forming connections: $\theta_{pre}$ scales the presynaptic bouton counts, $\theta_{post}$ scales the postsynaptic target density, and $\theta_{postAll}$ scales the normalizing feature (Fig 2D, left). The parametrized DSO rule for a presynaptic neuron $i$ and postsynaptic neuron $j$ positioned in a subvolume $k$ is then given by

$$\text{DSO}_{i,j,k}(\boldsymbol{\theta}) = \frac{pre_i^{\theta_{pre}} \cdot post_j^{\theta_{post}}}{postAll_k^{\theta_{postAll}}}, \tag{2}$$

(see Methods & materials for details).

The next step towards applying SBI is selecting measured data $x_o$ to constrain the rule parameters. We selected seven measurements of connection probabilities of neuronal

populations mapping from the ventral posterior medial nucleus (VPM) of the thalamus to different layers and cell types in the barrel cortex, as proposed by [19]: layer four (L4), layer four septum (L4SEP), layer four star-pyramidal cells (L4SP), and layer four spiny stellate cells (L4SS) [34], layer five slender-tufted intratelencephalic cells (L5IT), layer five thick-tufted pyramidal tract cells (L5PT), and layer six [35]. Thus, overall, one simulation of the wiring rule consisted of three steps: First, applying the rule with a given set of parameters to the structural features of every combination of neuron-pair-subvolume to obtain connection probabilities; second, sampling synapses from the Poisson distribution given the probabilities; and third, calculating the summary statistics matching the seven measured VPM-barrel cortex population connection probabilities (Fig 2A-D; see Methods & materials for details).

Our SBI approach can be summarized as follows: Given a generative model that can simulate barrel cortex connectomes from many different parametrizations of the DSO rule, we use SBI to infer those parametrizations that agree with connection probabilities measured in the barrel cortex. To perform SBI, we define a prior distribution $p(\theta)$ over the DSO rule parameters and generate training data for SBI by sampling random parameter values from the prior and simulating corresponding connection probabilities (Fig 2D, see Methods & materials for details). We selected the prior such that sampling random parameter values from the prior resulted in connection probabilities covering a broad range of values, including the measurements (Fig 2D, right, see Experimental settings and S1 Text for details). The prior also contained parameter combinations that led to extreme values, e.g., connection probabilities close to zero. SBI leverages this diverse set of parameter-data pairs to learn an approximation to the posterior distribution over DSO parameters $p(\theta|x_o)$ and thereby identifies those parameters that reproduce the measured data.

**SBI performs accurately on simulated data.**    Before applying SBI to infer the parameters of the DSO rule given measured data, we validated its accuracy on simulated data. As a first step, we considered a reduced version of the DSO rule simulator for which it was possible to obtain a ground-truth reference posterior distribution (see section Methods & materials for details). Using this reference solution, we checked whether SBI infers the posterior accurately and how many training simulations it requires. We compared three SBI algorithms: Sequential Neural Posterior Estimation (SNPE, [27–29]), which performs SBI sequentially over multiple rounds focusing inference on one particular observation; its non-sequential variant NPE; and a classical rejection-sampling-based approach called Sequential Monte Carlo (SMC, [38, 39]). We found that all three methods can accurately infer the reference posterior distribution (S1A and S1B Fig) but that they differ in terms of simulation efficiency, i.e., how many model simulations are required for accurate inference: SNPE was slightly more efficient than NPE, and both were substantially more efficient than SMC (S1B Fig).

As a second step, we performed two checks to validate SBI on the full version of the DSO rule simulator for which no reference solution was available. First, we used simulated-based calibration (SBC, [40]) to check whether the variances of posterior distributions inferred with SBI were well-calibrated, i.e., that the posteriors were neither too narrow (overconfident) nor too wide (conservative). We found that SNPE and NPE run with *simulated* observed data inferred well-calibrated posteriors for all three parameters (S1C Fig). Lastly, we checked the predictive performance of SBI by generating simulated data using parameter values sampled from the inferred posterior. We found that the predicted data resembles the (simulated) observed data (S1D Fig, see Methods & materials for details).

## SBI identifies many possible wiring rules and reveals parameter interactions

After evaluating SBI with simulated data, we applied it to infer the posterior over DSO rule parameters given the seven measured VPM-barrel cortex connection probabilities [34, 35]. Our analysis of the inferred posterior distribution revealed three key insights.

**The posterior identifies many data-compatible wiring rule configurations.** We found that the inferred posterior distribution was relatively broad, indicating that there are many parameter combinations with a high probability of explaining the measured data. The most likely parameter combination, i.e., the maximum of the one-dimensional posterior marginal distributions (Fig 3A, blue in diagonal subplots) was $\boldsymbol{\theta}_{MAP} = [1.14, 1.1, 1.08]$, corresponding to a relatively similar weighting of pre- and postsynaptic DSO rule features. This was in line with the assumption of the initial "a-priori" version of the DSO rule ($\boldsymbol{\theta} = [1, 1, 1]$, orange lines in Fig 3A). Importantly, however, the posterior marginal standard deviations of $\hat{\sigma}_{pre} = 0.2$, $\hat{\sigma}_{post} = 0.2$ and $\hat{\sigma}_{postAll} = 0.1$ showed that this is not the only solution but that there are several other parameter combinations with high posterior probability. For example, sampling parameter values from the posterior for $\theta_{post}$ would return values lying mostly in an interval as broad as [0.7, 1.5] (95% posterior credible interval). Despite this relatively broad range of plausible parameter values, we still found that the posterior predictive distribution, i.e., connection probabilities simulated with parameters sampled from the posterior, clustered around the measured data and closely matched the data simulated from the initial version of the DSO rule (Fig 3B). Quantitatively, we observed that all except the L4SS and L5IT measurements were located within one standard deviation of the posterior predictive distribution, which matched the standard deviation expected from the sample size used in the experiments (see Methods & materials for details, [34, 35]). How can so many parameter configurations from such broad ranges all result in similar data?

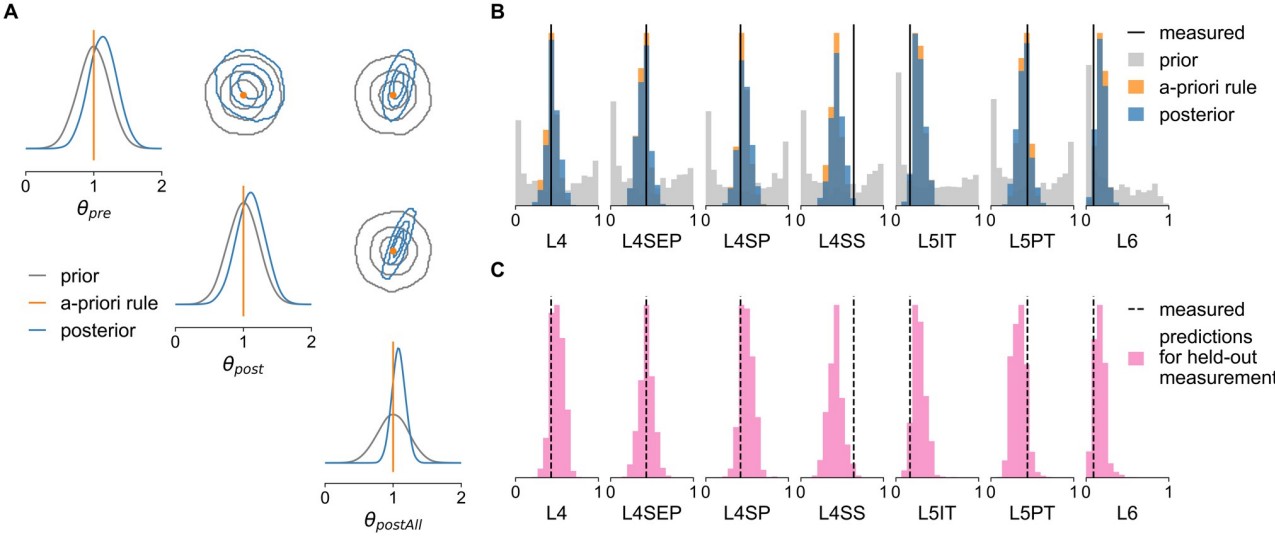

**Fig 3. SBI posterior reveals parameter interactions and predicts unseen data.** (**A**) The posterior over the three wiring rule parameters scaling the DSO features (inset) inferred with SBI (blue) and the initial prior distribution over parameters (gray). The corner plot shows the one-dimensional marginal distribution of each parameter on the diagonal and the pairwise two-dimensional marginals on the off-diagonal (contour lines show the 34%, 68%, and 95% credible regions). (**B**) Comparison of measured connection probabilities (black, [34, 35]) with those simulated with parameter values sampled from the inferred posterior (blue), from the prior (gray) and the unparametrized a-priori DSO rule (orange). (**C**) Each panel shows the predictions for one held-out measurement generated from a posterior that was trained and conditioned only on the other six measurements, i.e., each panel refers to a different posterior.

**Posterior analysis reveals biologically plausible parameter interactions.** Having access to the full posterior distribution and its covariance structure allowed us to answer this question. Inspection of the two-dimensional marginals of each parameter pair indicated a correlation structure substantially different from the uncorrelated prior distribution (Fig 3A, off-diagonal subplots). To quantify this, we estimated the Pearson correlation coefficients of 10, 000 parameter values sampled from the posterior distribution. We found a negative correlation between $\theta_{pre}$ and $\theta_{post}$ (Pearson correlation coefficient $\rho = -0.14$) and positive correlations between $\theta_{postAll}$ and $\theta_{pre}$ as well as $\theta_{post}$ ($\rho = 0.48$ and $\rho = 0.79$, respectively). These correlations are plausible given the design of the DSO rule (see Eq 2 and Fig 3A, inset). For example, the negative correlation between $\theta_{pre}$ and $\theta_{post}$ indicated that when increasing the value of $\theta_{pre}$, we would have to decrease $\theta_{post}$ in order to obtain the same overall number of connections for a particular cell type. This suggests that a stronger influence of presynaptic boutons on the connection probability requires a weaker influence of postsynaptic target targets on the connection probability.

The correlations further suggested that all three structural features are relevant in predicting the connection probabilities: Once one parameter is fixed, the values of the other parameters are strongly constrained. Having access to the full posterior distribution allowed us to quantify this by calculating the *conditional* correlations between the parameters. We obtained the conditional correlations by conditioning the posterior on one parameter dimension—i.e., holding it at a fixed value—and calculating the correlation between samples drawn from the resulting two-dimensional conditional posterior, once for each of the three parameters $\theta_{pre}$, $\theta_{post}$ and $\theta_{postAll}$ (see Methods & materials for details). The resulting correlation coefficients were substantially higher than without conditioning: -0.97 between $\theta_{pre}$ and $\theta_{post}$ and 0.99 between the other two parameter combinations (see S3 Fig for a visualization of the conditional posteriors). This result confirmed that while the overall range of data-compatible wiring rule parameters is relatively large (Fig 3A), once one parameter is fixed, the other two are constrained to a very small range of values. Furthermore, the strong conditional posterior correlations indicated that the DSO rule with three parameters is overparametrized, i.e., a parametrization of the DSO rule with only two parameters likely suffices to explain the measured data (see S2 Text for details).

Collectively, the SBI approach enabled us to efficiently identify several different DSO rule configurations that can all explain the measurements, among them the initially proposed version of the DSO rule. The analysis of the inferred posterior showed that the number of presynaptic boutons and the number of postsynaptic contact sites (and, by extension, axonal and dendritic path length) are sensitive and strongly interdependent structural features for predicting synaptic connectivity. Further, we found that the inferred parameters attribute similar weights to the DSO rule features and could predict the measurements well (except for the L4SS and L5IT cell types), indicating that multi-synaptic contacts at pre- or postsynaptic sites [36, 37] do not significantly impact the predictions of the DSO rule. The mismatch between the predicted and measured connection probabilities that we found for the L4SS and L5IT cell types (Fig 3B) suggests the DSO-rule assumption of similar scaling of DSO features does not hold here. Applying the SBI approach to an extension of the DSO rule with additional features or separate scaling factors for these cell types would likely reproduce these measurements more accurately.

**SBI posterior predicts unobserved connection probabilities.** To demonstrate the utility of SBI-enabled generative models as a tool for hypothesis generation, we investigated how one can make predictions on unobserved data. Above, we performed SBI to constrain the wiring rule parameters using only the seven measured connection probabilities. However, in principle, the structural model provides access to the entire (simulated) connectome of one barrel

cortex column. Thus, it allows us to make predictions about other features of the connectome that have not been measured yet. To test this approach, we repeated the SBI training procedure seven times, holding out each connectivity measurement once from the training data set, i.e., we trained the posterior estimator on pairs of parameters and data, $(\theta, x)$, where $x$ has six entries instead of seven. After training, we obtained seven different posteriors, each conditioned on six of the seven measured connection probabilities. We then sampled parameter values from every posterior, simulated the corresponding barrel cortex connectomes, and calculated *all seven* connection probabilities.

The posterior predictive distributions for held-out measurements clustered around the actual measurement values for all of the seven cell types, except for L4SS and L5IT (Fig 3C), and closely resembled the predictive distributions of the posterior inferred given all measurements (Fig 3B). To quantify this, we performed a classifier-based two-sample test (C2ST, [41, 42]): We trained a two-layer artificial neural network to classify the predictions of the posterior inferred using all measurements (class 1) from the predictions of the posterior inferred using only six measurements (class 2). In this setting, a classification accuracy close to 0.5 indicates performance at chance level, i.e., the two classes are indistinguishable. We found that the C2ST accuracy was below 0.55 for all measurements except for L4SS (0.68) and L5PT (0.59). This indicated that for all but the L4SS and L5PT cell types, the weighting of the DSO features could be accurately predicted from the other cell types, in line with the results presented above. Collectively, these cross-validation results suggested that the structural model paired with the SBI-enabled wiring rule enables us to make experimentally testable predictions. For example, one could predict connection probabilities of cell types different from the seven measured here or other connectivity features of the rat barrel cortex available in the structural model (see below).

## Using SBI to rule out invalid wiring hypotheses

Above, we demonstrated that SBI provides a quantitative way to identify valid wiring rule configurations from a large set of hypothesized wiring rules. SBI can also be used to rule out invalid hypotheses, e.g., to show that an existing hypothesis does not agree with empirical data. One such debated hypothesis in connectomics is the so-called *Peters' rule* [43, 44]. According to this hypothesis, neurons form connections whenever their axons and dendrites are in close proximity, i.e., Peters' rule can be formulated as "axo-dendritic proximity predicts connectivity" [19]. However, several empirical and theoretical studies on rat, mouse, and human connectomes found substantial evidence against Peters' rule [19, 45–47].

Here, we show that SBI provides an alternative, quantitative way to discard this hypothesis for the rat barrel cortex. We formulated two wiring rules that implement the proximity hypothesis in the structural model of the barrel cortex at two spatial scales: one predicting connections on the neuron-to-neuron level (Fig 4) and one predicting synapse counts at the subcellular level (Fig 5). Both wiring rules have one free parameter, i.e., they incorporate many different proximity hypotheses, but there is one particular parameter value corresponding to Peters' rule. We used SBI to infer the posterior distribution over the rule parameters given the seven measured VPM-barrel cortex connection probabilities. Subsequently, we compared inferred parameter values and their predictions with those corresponding to Peters' rule.

**Neuron-level rule.** At the neuron-to-neuron level, we defined the proximity of two neurons as the number of subvolumes $v$ they share in the structural model and introduced a threshold parameter acting on the proximity: Neurons $i$ and $j$ form a connection $c_{i,j}$ if the number of subvolumes $v_{ij}$ that contain presynaptic structures of neuron $i$ *and* postsynaptic

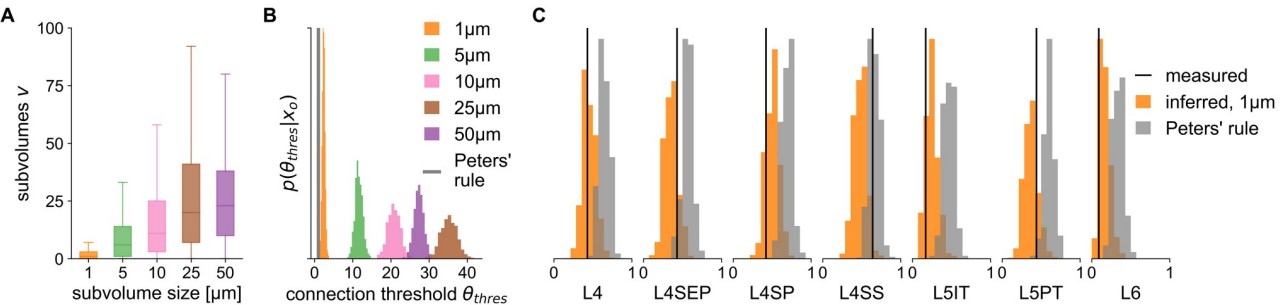

**Fig 4. Neuron-level wiring rule inferred with SBI differs from Peters' rule.** We used SBI to infer a proximity-based wiring rule at different spatial resolutions of the rat barrel cortex model and compared its predictions to that of Peters' rule. **(A)** Distributions of the shared subvolumes $v$ between neurons in the barrel cortex model for each spatial resolution (subvolume edge length, see legend in (b)). **(B)** SBI posteriors inferred over the connection threshold parameter of the wiring rule ($\theta_{thres}$, number of shared subvolumes required to form a connection), shown for each spatial resolution (colors), and for Peters' rule assuming $\theta_{thres} = 1$ (gray). **(C)** Connection probabilities simulated from the inferred posterior (orange) and Peters' rule (gray) compared to the measured connection probabilities (black).

structures of neuron $j$, exceeds a threshold parameter $\theta_{thres}$:

$$c_{i,j}(\theta_{thres}) = 1 \text{ if } v_{ij} > \theta_{thres} \text{ else } 0. \tag{3}$$

To compare the resulting pair-wise connections between neurons in the barrel cortex model to the measured connection probabilities, we mapped them to the corresponding population connection probabilities as described above (see section Methods & materials for details).

The structural feature used in this rule is the number of shared subvolumes between two neurons (in contrast to the *subcellular* features used in the DSO rule above). This feature directly depends on the spatial resolution of the structural model, i.e., the edge length of the subvolume used to construct the model. Therefore, we calculated the structural features at five different spatial resolutions: 50, 25, 10, 5, and 1 μm. We found that the overall number of shared subvolumes among neurons increased with edge length, reflecting the increase in subvolume size (Fig 4A).

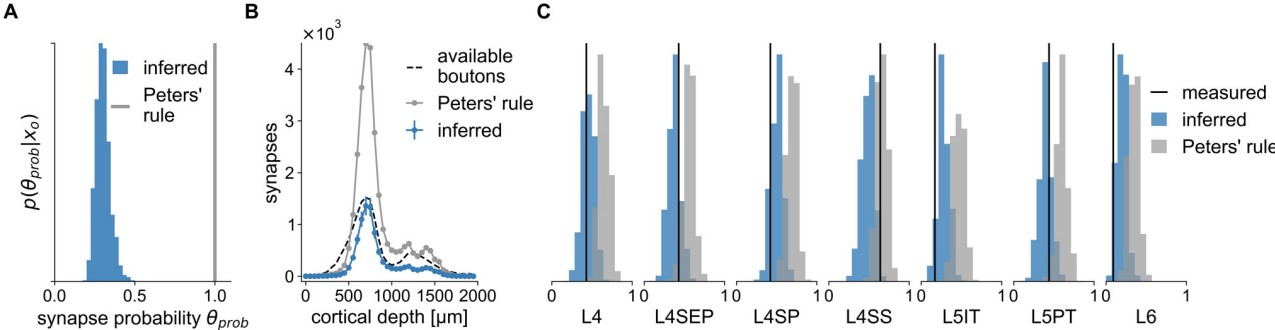

**Fig 5. Synapse-level wiring rule inferred with SBI differs from Peters' rule.** We compared an SBI-inferred parametrized wiring rule predicting synapse counts on the subcellular level with a corresponding formulation of Peters' rule. **(A)** SBI posterior for the wiring rule parameter $\theta$ (probability of forming a synapse if two neurons are close), compared to Peters' rule assuming $\theta = 1$ (gray). **(B)** Number of synapses predicted by the inferred posterior (blue) and Peters' rule (gray) compared to the number of presynaptic boutons realistically available in the structural model (dashed black), plotted over the entire cortical depth of the barrel cortex column. **(C)** Connection probabilities simulated from the inferred synapse level posterior (blue) and Peters' rule (gray) compared to the measured connection probabilities (black).

When applying Peters' rule to the barrel cortex model at the neuron level, a connection occurs whenever two neurons share at least one subvolume, i.e., the connection threshold would be $\theta_{thres} = 1$. Does this assumption hold for the barrel cortex at the neuron-to-neuron level as well? To answer this question quantitatively, we used SBI to infer the threshold parameter $\theta_{thres}$ of the neuron level rule for each edge length. We observed that the inferred threshold parameters shifted to larger values with increasing edge length, e.g., the higher the spatial resolution, the fewer common subvolumes were required to obtain a connection (Fig 4B). This was in line with our observation that the overall number of shared subvolumes available in the structural model increased with increasing edge length (Fig 4A). However, irrespective of the spatial resolution, all inferred threshold parameters were substantially larger than the $\theta_{thres} = 1$ of Peters' rule, reaching from $\theta_{thres} \approx 3$ (posterior mean) for the 1 μm-subvolume model (Fig 4B, orange) to $\theta_{thres} \approx 28$ for the 50 μm-subvolume model (Fig 4B, violet). Accordingly, the comparison of predictive performances showed that for all but the L4SS cell type, the difference between measurement and the average prediction was smaller for the inferred rule than for Peters' rule and below one standard deviation expected in the experiments (Fig 4C). For the L4SS cell type, we found that data simulated with Peters' rule was close to the measurement (0.02 difference).

**Synapse-level rule.**   We repeated this test of Peters' rule at the subcellular level as well. Here, we defined a probabilistic rule: Whenever a presynaptic structure of neuron $i$ and a post-synaptic structure of neuron $j$ are present within the same subvolume $k$, they form a synapse with probability $\theta_{prob}$:

$$c_{i,j,k}(\theta_{prob}) \sim \text{Bernoulli}(\theta_{prob}) \text{ if axon of } i \text{ and dendrite of } j \text{ are present in } k. \qquad (4)$$

This rule predicts synapses for every neuron-pair-subvolume combination using the structural model with subvolumes of 1 μm edge length. To compare the simulated synapse counts to the measured connection probabilities, we calculated simulated connection probabilities as described above. The posterior distribution over the connection probability parameter $\theta_{prob}$ inferred with SBI centered around $\theta_{prob} \approx 0.3$ (Fig 5A). This result suggests that in only thirty percent of the locations where axon and dendrite are close to each other (shared 1 μm subvolume), the rule predicts a synapse, which is substantially lower than the value of $\theta_{prob} = 1$ corresponding to Peters' rule. Accordingly, simulating parameter values sampled from the posterior resulted in connection probabilities closer to the measured ones than those predicted by Peters' rule (smaller difference between average prediction and measurement, Fig 5C), except for the L4SS cell type.

For another comparison of Peters' rule with the inferred wiring rule at the synapse level, we leveraged the predictive properties of the structural model and the SBI posterior. In particular, the structural model provides access to estimates of the number of biologically available boutons across the cortical depth of the barrel cortex column [19]. The SBI posterior allowed us to simulate data according to the inferred wiring rule parameters. Thus, it was possible to compare the estimate of the number of empirically available boutons of each presynaptic VPM neuron with the number of simulated synapses from the inferred wiring rule and Peters' rule. We found that the inferred rule predicted synapses close to or below the total number of available boutons (Fig 5B), in contrast to Peters' rule, which predicted more synapses than biologically plausible.

Our results demonstrate how SBI can be applied to competing wiring hypotheses to quantitatively rule out a specific invalid hypothesis: One incorporates the hypothesis into a parametrized model and compares the SBI-inferred parameters to those corresponding to the hypothesis. In case of Peters' rule, the SBI analysis confirmed that axo-dendritic proximity

alone cannot predict connectivity observed empirically in the rat barrel cortex—it predicted too high connection probabilities for all measured cell types except L4SS (Fig 5C) and consistently too many synapses across all cortical layers (Fig 5B). This is in line with previous results showing that the number of dendrites and axons close to each other exceeds the number of synapses by 1–2 orders of magnitude [19]. Thus, we can conclude that to explain connectivity in the rat barrel cortex, wiring rules cannot be based solely on axo-dendritic proximity. They also have to take into account subcellular features like pre- and post-synaptic structures along axons and dendrites.

## Discussion

What principles are behind the complex connectivity patterns of neural networks that shape brain function? Connectomics aims to answer this question by acquiring detailed data about the structural and functional composition of the brain. Over the last few years, the development of new computational approaches for analyzing the resulting large amounts of data and testing the derived hypotheses gained momentum [11, 48]. One computational approach is to leverage generative models for testing hypotheses about the connectome, e.g., to implement a hypothesized wiring rule in a computational model and ask whether model simulations can reproduce the measured connectivity patterns of a specific brain region [13]. However, identifying the free parameters in the wiring rule that reproduce measured connectivity data can be challenging.

We introduced a method that renders the generative modeling approach to connectomics more efficient, enabling us to systematically infer *all* data-compatible parameters of a given computational model. Instead of manually refining a specific generative model to match the data, we equipped it with parameters such that it represents several candidate hypotheses. We then used Bayesian parameter inference to infer the posterior distribution over model parameters conditioned on the measured data. The inferred distribution represents all candidate parameter configurations, i.e., hypotheses, capable of reproducing the measured data. By relying on simulation-based inference (SBI) methods that do not require access to the likelihood function of the model, we were able to apply our approach to the simulation-based generative models commonly used in computational connectomics.

To demonstrate the utility of this approach, we employed it to constrain several wiring rules —at different spatial scales—with connectivity measurements from the rat barrel cortex. We first showed that the inference method is accurate in a scenario with a ground-truth reference solution. Next, in the realistic setting with measured connectivity data, we retrieved many different wiring rule configurations that could explain the measured data equally well. Analyzing the geometrical structure of the inferred posterior distribution revealed strong correlations between wiring rule parameters that are in line with their biological interpretations. Importantly, we were able to accurately predict held-out connectivity measurements, demonstrating the method's utility in making experimentally testable predictions. Finally, we used our approach to quantitatively show that a wiring rule based solely on axo-dendritic proximity cannot explain barrel cortex connectivity measurements. Overall, these results demonstrate the potential benefits of the Bayesian inference approach, i.e., having access to the full posterior distribution over model parameters rather than manually optimizing for individual parameters one hypothesis at a time and the flexibility of SBI in requiring only simulated data to perform inference.

### Related work

The problem of identifying parameters of computational models that reproduce experimentally observed data has been addressed in computational connectomics before. For

example, [15] built a model of functional connections between brain regions and used optimization methods to find single best-fitting parameters capturing the functional MRI data measured in humans. [49] and [16] used Monte Carlo sampling methods for optimizing the parameters of synthetic networks of structural connectivity to match the topological properties of human connectomes recorded with MRI. Recently, [24] proposed a parameter estimation method for connectome generative models based on averaging over ensembles of best-fitting parameters. In contrast to our approach, these studies do not perform Bayesian inference but rely on optimization techniques that identify best-fitting solutions.

Aside from the above examples, computational models in connectomics are often implemented as complicated computer simulations that can generate simulated data but for which the underlying likelihood functions are not accessible, thus limiting our ability to perform Bayesian inference. To account for this limitation, we employed simulation-based inference (SBI, [25]) methods which only require simulations from the model to perform Bayesian inference. SBI has been applied previously in various fields, ranging from genomics [50], evolutionary biology [51, 52], computational and cognitive neuroscience [26, 53–57], to robotics [58], global health [59] and astrophysics [60, 61]. For the wiring rule examples presented here, we used sequential neural posterior estimation (SNPE, [27–29]), which performs neural-network-based conditional density estimation to estimate the posterior distribution from simulated data. Neural-network-based SBI approaches build on recent advances in probabilistic machine learning [27, 62, 63] and exhibit several advantages compared to more classical rejection-sampling-based techniques, commonly termed Approximate Bayesian Computation (ABC, [64]). For example, in contrast to ABC approaches, they do not require selecting a criterion for quantitatively comparing simulated with measured data. Furthermore, they can leverage the ability of neural networks to exploit continuities in the parameter space or automatically learn informative summary features from high-dimensional raw data. As a consequence, they are often more simulation-efficient (S1B Fig) and scale better to problems with more model parameters and high-dimensional data [42].

Bayesian inference methods have been employed in computational connectomics before, but in different ways, than the SBI approach proposed here. [65] built a probabilistic model of cell type-dependent connectivity in the mouse retina and proposed a non-parametric Bayesian algorithm that automatically predicts cell types and microcircuitry from connectomics data. More closely related to our approach, [66] performed Bayesian model comparison using a rejection-sampling approach [67] to compare a set of competing local circuit models in layer 4 of the mouse primary somatosensory cortex based on structural connectomics data. However, in contrast to our approach that infers the posterior distribution *over parameters* of one particular model, [66] inferred the posterior probabilities *over several models* (with the parameters of individual models integrated out). Moreover, their approach relied on classical rejection-sampling techniques that are less simulation-efficient compared to the neural-network-based SBI we employed and will likely not scale to higher-dimensional inference problems (S1B Fig, see [42] for a detailed comparison). Yet, there have been recent advances in neural network-based Bayesian model comparison [68–70] and simultaneous model and parameter inference using SBI [71]. Thus, a promising direction for future research in connectomics would be combining Bayesian model comparison and parameter inference, e.g., to compare competing wiring rule hypotheses on the basis of empirical data: One would simulate data from each wiring rule and use SBI to approximate the discrete posterior distribution over different wiring rules to select the wiring rule with the highest posterior probability and simultaneously use SBI to constrain the parameters of the selected rule (as proposed here).

## Applicability and limitations

In this work, we applied SBI to the specific problem of inferring wiring rules that can generate observed connectivity data. A key advantage of SBI is that it is generally applicable to any computational model capable of simulating data from a set of parameters, i.e., only data simulated from the model are required to infer the posterior distribution over model parameters. For example, one could make the algorithms for generating synthetic functional connectomes proposed by [15] or [49] amenable to SBI by introducing parameters of interest and then use SBI to efficiently identify all data-compatible parameter values. Similarly, SBI could be applied to infer unknown parameters of iterative growth-based generative models as used in the works by [16, 24, 72]. Concretely, one would generate many different synthetic networks from a given growth model using different candidate parameter settings and then use the simulated data in the SBI framework to obtain the posterior distribution given empirical network data.

Posterior-targeting SBI approaches, like the NPE algorithm we used, have the advantage that they can obtain the posterior distribution for new data points without retraining the underlying artificial neural networks, i.e., they perform amortized inference [26, 27]. Another advantage of SBI is that it can leverage the ability of neural networks to automatically learn informative summary features from observations [26, 28, 29, 73, 74]. While we did not exploit this feature for the low-dimensional measured data in the wiring rule examples presented here, we believe that it will be essential for future applications of SBI in computational connectomics, e.g., when dealing with high-dimensional dense reconstructions of electron-microscopy data [30–32].

SBI's dependency on simulated data and neural network training also entails several limitations: the inferred posterior distributions are only approximations of the unknown actual posterior distribution. Therefore, applying SBI requires careful evaluation of every problem at hand. In theory, SBI does recover the unknown posterior distribution when given enough training data [27]. In practice, however, the accuracy and reliability of SBI strongly depend on the complexity of the inference problem, e.g., on the number of model parameters, the complexity of the data, and the simulation time. Previous studies have successfully applied SBI in scenarios where the simulator has a runtime on the order of seconds and up to thirty parameters [26, 54, 74], but these numbers strongly depend on the problem and available computational resources.

Another limitation of SBI is the problem of model misspecification. SBI generally assumes that the generative model is well-specified, i.e., that it can simulate data that is very similar to the measured data. If this is not the case, then the inferred posterior can be substantially biased [75, 76]. We recommend performing prior predictive checks to detect model misspecification, i.e., generating a large set of simulated data with parameters sampled from the prior distribution and checking whether the measured data lies inside the distribution of simulated data, as demonstrated in the wiring rule example (Fig 2D, see S1 Text for details). Recent methodological work in SBI addresses this problem, e.g., by automatically detecting model misspecification [77] or by explicitly incorporating the model mismatch into the generative model [78].

More generally, applying SBI to new inference problems requires several choices by the practitioner, from prior predictive checks and model-checking to selecting suitable neural network architectures and validating the inferred posterior distribution. As a general guideline, we recommend following the steps we performed for the wiring rule example: First, we investigated the accuracy of SBI and estimated the required number of training simulations by testing it in a scenario with a known reference solution. Second, we ensured that the inferred posterior distribution has well-calibrated uncertainty estimates using simulation-based calibration [40]. Third, we checked whether the parameter values identified by SBI accurately reproduced the

measured data (see Methods & materials for details). Additionally, we recommend guiding hyperparameter choices by resorting to well-tested heuristics and default settings available in open-source software packages developed and maintained by the community. We performed all our experiments, evaluation steps, and visualization using the sbi toolkit [33].

## Conclusion

We present SBI as a method for constraining the parameters of generative models in computational connectomics with measured connectivity data. The key idea of our approach is to initially define a probability distribution over many possible model parameters and then use Bayesian parameter inference to identify all those parameter values that reproduce the measured data. We thereby replace the iterative refinement of individual model configurations with the systematic inference of all data-compatible solutions. Our approach will be applicable to many generative modeling scenarios in computational connectomics, providing researchers with a quantitative tool to evaluate and explore hypotheses about the connectome.

## Methods & materials

### Bayesian inference for computational connectomics

We introduced Bayesian inference as a tool to identify model parameters of generative models in computational connectomics, given experimentally observed data. Bayesian inference takes a probabilistic view and defines the model parameters and data as random variables. It aims to infer the conditional probability distribution of the model parameters conditioned on the observed data, i.e., the *posterior distribution*. Bayes' rule defines the posterior distribution as

$$p(\boldsymbol{\theta}|x_{\mathrm{obs}}) = \frac{p(x_{\mathrm{obs}}|\boldsymbol{\theta})\,p(\boldsymbol{\theta})}{p(x_{\mathrm{obs}})}, \tag{5}$$

where $p(x_{\mathrm{obs}}|\boldsymbol{\theta})$ is the likelihood of the data given model parameters, $p(\boldsymbol{\theta})$ is the prior distribution over model parameters and $p(x) = \int_{\boldsymbol{\theta}} p(x|\boldsymbol{\theta})p(\boldsymbol{\theta})d\boldsymbol{\theta}$ is the so-called *evidence*. Thus, performing Bayesian inference requires three components:

1. Experimentally observed data $x_{obs}$.

2. A *likelihood* $p(x_{obs}|\boldsymbol{\theta})$, which defines the relationship between model parameters and data. In our setting, the likelihood is implicitly defined by the computational model, i.e., by the simulator generating connectomics data $x$ given model parameters $\boldsymbol{\theta}$. The simulator needs to be stochastic, i.e., when repeatedly executed with a fixed parameter $\boldsymbol{\theta}$, it should generate varying data. Technically, given a fixed parameter value $\theta$, the likelihood defines a probability distribution over $x$, and simulating data corresponds to sampling $x \sim p(x|\boldsymbol{\theta})$.

3. A *prior* distribution $p(\boldsymbol{\theta})$. The prior incorporates prior knowledge about the parameters $\boldsymbol{\theta}$, e.g., biologically plausible parameter ranges or known parameter correlations.

The posterior distribution $p(\boldsymbol{\theta}|x_{obs})$ inferred through Bayes' rule characterizes all model parameters likely to reproduce the observed data. For example, model parameters with a high probability under the posterior distribution will result in data close to the observed data. In contrast, parameters from low probability density regions will likely generate data different from the observed data.

In most practical applications, it is hard to obtain an analytical solution to Bayes' rule because the evidence $p(x) = \int p(x|\theta)p(\theta)d\theta$ is challenging to calculate. There exists a large set of methods to perform approximate inference, e.g., Markov Chain Monte Carlo sampling

(MCMC, [79, 80]). MCMC methods can be used to obtain samples from the posterior distribution. However, they require evaluation of the likelihood function of the model, and computational models in connectomics are usually defined as scientific simulators for which no analytical form of the underlying likelihood is available or numerical approximations are computationally expensive.

**Simulation-based inference.** Simulation-based inference (SBI, [25]) allows us to perform Bayesian inference without numerical evaluation of the likelihood by requiring only access to simulations from the model. The idea of SBI is to generate a large set of pairs of model parameters and corresponding simulated data and use it as training data for artificial neural networks (ANN). The employed ANNs are designed to approximate complex probability distributions. Thus, they can be used to approximate the likelihood to then obtain posterior samples via MCMC [81–84] or the posterior distribution directly [27–29]. Once trained, the neural networks are applied to the experimentally observed data to obtain the approximate posterior. In our work, we used an SBI approach called *sequential neural posterior estimation* (SNPE, [27, 29]).

**Neural posterior estimation.** Neural posterior estimation (NPE) uses an artificial neural network $F(x)$ to learn an approximation of the posterior from training data pairs $\{(\boldsymbol{\theta}_i, x_i)\}_{i=1}^N$, where $\boldsymbol{\theta}$ is sampled from a prior $\boldsymbol{\theta}_i \sim p(\boldsymbol{\theta})$, and $x$ is simulated from the model $x_i \sim$ simulator $(\boldsymbol{\theta}_i)$. The density estimator $F(x)$ is trained to construct a distribution that directly approximates the *posterior*. It is usually defined as a parametric family $q_\phi$ with parameters $\phi$, e.g., a mixture density network (MDN, [85]), or a normalizing flow [63]. For example, suppose $q$ is a mixture of Gaussians, then $F$ would take the data as input and predict the parameters $\phi$, $\phi = F(x)$, where $\phi$ contains the means, the covariance matrix, and the mixture weight of each mixture component. $F(x)$ is trained to predict the parameters $\phi$ from $x$ by minimizing

$$-\frac{1}{N} \sum_{i=1}^N \log q_{\phi=F(x_i)}(\boldsymbol{\theta}_i | x_o).$$

This training loss implicitly minimizes the Kullback-Leibler divergence between the true posterior and the approximation $q_\phi(\boldsymbol{\theta}|x)$. It will converge to zero, i.e., NPE will infer the true posterior, in the limit of infinite training data and given density estimator that is flexible enough [27, 62]. Algorithm 1 summarizes the algorithmic steps of NPE; see [29] for details.

Once NPE is trained on simulated data, it can be applied to the actual observed data $x_{\text{obs}}$, e.g., $\phi = F(x_{\text{obs}})$, to obtain an approximation to the desired posterior:

$$q_\phi(\boldsymbol{\theta}|x_{\text{obs}}) \approx p(\boldsymbol{\theta}|x_{\text{obs}}). \tag{6}$$

Importantly, NPE applies to any newly observed data without retraining the density estimator, i.e., the inference with NPE is *amortized*. There is also a sequential variant of NPE called SNPE, where the training is performed over several rounds to focus the density estimator on a specific observation $x_{\text{obs}}$. In each new round of SNPE, the new training data is not generated with parameters sampled from the prior but from the posterior estimate of the previous round. While the sequential approach can be substantially more sampling-efficient compared to NPE, i.e., requiring fewer training simulations to obtain a good posterior approximation for a given $x_{\text{obs}}$ [42], it comes with two caveats. First, it requires retraining for every new $x_{\text{obs}}$. Second, using a proposal distribution different from the prior for simulating new data requires a correction, resulting in additional algorithmic choices and challenges. Over the last few years, different approaches have been proposed to perform this correction [27–29, 86]. We used SNPE with the correction proposed by [29].

**Algorithm 1**: Single round Neural Posterior Estimation as in [27]

```
input simulator p(x|θ), prior p(θ), observed data x_obs
for j = 1 : N do
  Sample θ_i ~ p(θ)
  Simulate x_i ~ p(x|θ_i)
end
x ← argmin −1/N ∑_i^N log q_{F(x_i,φ)}(θ_i)
Set p̂(θ|x_o) = q_{F(x_obs,φ)}(θ)
return Samples from p̂(θ|x_obs); density estimator q_{F(x,φ)} (θ)
```

**Posterior validation.** In theory and with unlimited training data, NPE will converge to the true (unknown) posterior distribution. However, training data is limited in practice, and the underlying posteriors can be high-dimensional and complex. Thus, it is essential to validate the approximate posterior. There are two common techniques for validating SBI even in the absence of a reference posterior: predictive checks [87] and calibration checks, e.g., simulation-based calibration (SBC, [40, 88]).

**Predictive checks.** Predictive checks can be applied to either the prior or the posterior. The *prior predictive check* is applied before the inference. It checks whether the model can produce data close to the experimentally observed data $x_{obs}$, i.e., that the distribution obtained by sampling from the prior and simulating the corresponding data contains $x_{obs}$. If this is not the case, the model or the prior could be misspecified and should be refined before applying SBI. We performed the prior predictive check for the DSO rule simulator by sampling 100,000 parameters from the prior and ensuring that the resulting distribution of simulated connection probabilities covers the seven measured values (see S1 Text for details).

The posterior predictive check tests the predictive performance of the posterior. It should be applied after the inference by simulating data using parameters sampled from the posterior:

$$x_p \sim \text{simulator}(\theta_p) \text{ where } \theta_p \sim p(\theta|x_{obs}).$$

The simulated data should cluster around the observed data with a variance on the order of the variance expected from the simulator. We performed this check for all inferred wiring rules by simulating 1,000 data points using 1,000 parameters sampled from the corresponding SBI posterior.

**Simulation-based calibration.** The variance of the posterior distribution expresses the uncertainty in the parameters. Simulation-based calibration (SBC) provides a way to check whether these uncertainties are, on average, well-calibrated, i.e., that the posterior is (on average) neither too broad (under-confident) nor too narrow (over-confident). The basic idea of SBC is the following. Suppose one uses an SBI method to obtain $i = 1, \ldots, N$ different posteriors $p(\theta|x_i)$ for different observations $x_i$ generated from different parameters $\theta_i$ sampled from the prior. If one determines the rank of each parameter $\theta_i$ among samples from its corresponding posterior $p(\theta|x_i)$, then the posteriors obtained with this SBI method have well-calibrated uncertainties if the collection of all $N$ ranks follows a uniform distribution [40]. To check whether the posterior obtained with SBI is well-calibrated, we repeated the inference with NPE $N = 1000$ times (no retraining required), using data generated from the simulator with parameters sampled from the prior. Subsequently, we performed a visual check for uniformity of the corresponding SBC ranks by comparing their empirical cumulative density function against that of a uniform distribution.

## A generative structural model of the rat barrel cortex

We demonstrated the utility of SBI for computational connectomics by constraining wiring rules in a structural model of the rat barrel cortex with connectivity measurements. To fulfill the prerequisites of Bayesian inference defined above, we set up a simulation-based model for

simulating wiring rules in the barrel cortex model. The wiring rule simulator has three components:

1. a *structural model* that provides features (Fig 2A),

2. a parametrized *wiring rule* that is applied to the features to simulate a connectome (Fig 2B),

3. calculation of *summary statistics* from the simulated connectome to match the available measurements (Fig 2C).

**The structural model.**   The structural model is a digital reconstruction of the rat barrel cortex constructed from detailed measurements of cell types and their morphologies, locations, and sub-cellular features, including single boutons and dendrites, obtained from several animals [19–23, 89]. These measurements and their digital reconstructions were copied and arranged according to measured cell type distributions in all cortical layers to obtain a realistic estimate of the structural composition of a large part of the entire barrel cortex. The model contains around 477,000 excitatory and 77,000 inhibitory neurons, resulting in more than 5.5 billion synaptic sites. Furthermore, it incorporates the projections from the ventral posterior medial nucleus (VPM) of the thalamus to all cortical layers. The model gives access to several cellular and subcellular structural features, including presynaptic boutons and postsynaptic target counts (spine densities). These features were collected for every neuron segment in every subvolume of the model. The model does not contain synaptic connections but only the structural features. Thus, it allows simulating the effect of different wiring rules by applying the rule to the structural features and comparing the resulting connectome to experimental measurements [19].

**Dense structural overlap wiring rule.**   We applied a wiring rule to the structural model to turn it into a simulation-based model that can generate simulated connectomes of the rat barrel cortex. As a wiring rule, we used the dense structural overlap (DSO) rule introduced by [19, 23], which proposes that two neurons form a synapse depending on their locally available structural subcellular features summarized as DSO. The DSO is the product of the numbers of pre- and postsynaptic structures, *pre* and *post*, that a presynaptic neuron *i* and a postsynaptic neuron *j* contribute to a subvolume *k* relative to the total number of postsynaptic structures contributed by all neurons, *postAll* (Fig 2B):

$$\mathrm{DSO}_{i,j,k} = \frac{pre_i \cdot post_j}{postAll_k}. \tag{7}$$

We assumed that the number of connections between any neuron pair (*i,j*) within a subvolume *k* is given by a Poisson distribution with the DSO as the rate parameter (Fig 2C, [23]):

$$c_{ijk} \sim \mathrm{Poisson}(\mathrm{DSO}_{i,j,k}).$$

The DSO rule is stochastic, i.e., it samples different synapse counts every time it is applied to the structural model. However, the contribution of each structural feature to the DSO is fixed. To allow for more flexibility in the relative weighting of each feature, we generalized the DSO rule by introducing three scaling parameters for the three structural features, $\theta_{pre}$, $\theta_{post}$, and $\theta_{postAll}$:

$$\mathrm{DSO}_{i,j,k}(\boldsymbol{\theta}) \quad = \frac{pre_i^{\theta_{pre}} \cdot post_j^{\theta_{post}}}{postAll_k^{\theta_{postAll}}}. \tag{8}$$

We rewrote the parametrized rule as a Poisson *generalized linear model* (GLM, [90]) by transforming the features to the logarithmic space, stacking them as column vectors in a feature matrix $X_{ijk} = [\log pre(i, k); \log post(j, k); -\log postAll(k)]$ and arranging the scaling parameters in a vector $\boldsymbol{\theta}$:

$$
\begin{aligned}
\mathrm{DSO}_{i,j,k}(\boldsymbol{\theta}) \quad &= \exp\left(\log\left(\frac{pre_i^{\theta_{pre}} \cdot post_j^{\theta_{post}}}{postAll_k^{\theta_{postAll}}}\right)\right) \\
&= \exp(\theta_{pre} \log pre_i + \theta_{post} \log post_j - \theta_{postAll} \log post All_k) \\
&= \exp(\boldsymbol{\theta}^\top X_{ijk}) \\
c_{ijk}(\boldsymbol{\theta}) \quad &\sim \mathrm{Poisson}(\exp(\boldsymbol{\theta}^\top X_{ijk})).
\end{aligned}
$$

The generalized version of the DSO rule takes parameter combination $\boldsymbol{\theta}$ and generates simulated connectomes in the format of synapse counts. Each new parameter setting corresponds to a different variant of the DSO rule that could have generated the measured data. Note that setting $\boldsymbol{\theta} = [1, 1, 1]^\top$ would result in the expression introduced in Eqs 7 and 8.

**Mapping from simulated connectomes to measured connectivity data.** The generalized DSO rule and the chosen prior distribution jointly define a space of simulated connectomes associated with the DSO rule. The last step for constructing a simulation-based model is to map the simulated connectomes to the type of data that can be measured empirically. The measurements available for the barrel cortex are connection probabilities estimated from pairwise recordings between neurons in the ventral posterior medial nucleus (VPM) of the thalamus and different layers and cell types in the cortex: to layer 4 (L4), layer 4 septum (L4SEP), layer 4 star pyramidal cells (L4SP), and layer 4 spiny stellate cells (L4SS) [34], layer 5 slender-tufted intratelencephalic cells (L5IT), layer 5 thick-tufted pyramidal tract cells (L5PT), and layer 6 [35].

While the simulated connectome provided access to all neuron-pair-subvolume combinations in the structural model, the measurements were given only as estimated connection probabilities for different cell types. To calculate these connection probabilities from the simulated connectome, we identified the pairs from the presynaptic and postsynaptic neuron populations used in the experiments and calculated the connection probabilities from the corresponding simulated synapse counts. The number of probed neuron pairs was relatively small in the experiments, e.g., around 50 [34]. We tried to mimic this experimental setting in the simulation by selecting a random sample of 50 pairs from the thousands of possible pairs available in the simulated connectome. We then checked how many of those neuron pairs connected with at least one synapse for each of the seven populations. Algorithm 2 summarizes the steps for calculating the summary statistics.

**Algorithm 2**: Wiring rule simulator and summary statistics calculation
```
input structural model S, wiring rule R(θ), parameter θ ∼ p(θ), mea-
sured connectivity data x_obs
Simulate:
generate synapse counts for each neuron-pair i, j in each subvolume k:
c_ijk ∼ simulator(S,R,θ)
Summarize:
for each population m measured in x_obs do
  Find index set of neuron pairs Π_m belonging to population m
  Sample 50 random neuron-pair indices π ∼ Π_m
  Estimate population connection probability as average over connected
pairs: ∑_(i,j)∈π I(i→j)/|π|
end
```

```
return simulated connectivity data x
```

Overall, this provided us with a setup to perform Bayesian inference as defined above, to constrain the parameters of a hypothesized wiring rule in the rat barrel cortex:

1. observed data $x_{obs}$ given by seven measured connection probabilities between VPM and barrel cortex

2. a stochastic simulation-based model that generates simulated data $x$ according to the hypothesized DSO rule, given parameters $\boldsymbol{\theta}$

3. a prior over model parameters $\boldsymbol{\theta}$

This setup readily extends to other observed data, other simulation-based models, or priors.

### Experimental settings

**Simulation and SBI settings.**   For performing SBI on the DSO rule parameters, we used a Gaussian prior over the three parameters:

$$\boldsymbol{\theta} \sim \mathcal{N}(\mu_0 = [1, 1, 1]^\top, \Sigma_0 = 0.05\, I). \tag{9}$$

We chose the variance of the prior such that the distribution of simulated connection probabilities covered the range [0, 1] densely (see S1 Text for details).

For the inference on the distance-based wiring rules, we used a uniform prior over the shared subvolume threshold parameter $\theta_{thres}$ of the neuron-level rule

$$\theta_{thres} \sim \mathcal{U}(0, 100),$$

and a Beta distribution prior over the synapse probability parameter $\theta_{prob}$ of the synapse level rule

$$\theta_{prob} \sim \text{Beta}(\alpha = 2, \beta = 2).$$

The settings for generating training simulations for SBI were the same for all results: We drew 1,000,000 parameter values from the prior and simulated the corresponding synapse counts or neuron-level connections, followed by the summary step (except for the simulator used for benchmarking, see below).

To perform SBI, we used (S)NPE with Neural Spline Flows (NSF, [91]) as density estimator. The NSF hyperparameters were: five transforms, two residual blocks of 50 hidden units each, ReLU non-linearity, and ten spline bins, all as implemented in the public *sbi* toolbox [33]. The training was performed with a training batch size of 1, 000, a validation set proportion of 10%, and a convergence criterion of 20 epochs without improvement of validation loss. We used the different versions of NPE or SNPE as follows: For the benchmark of SBI against the MCMC reference posterior, we compared the non-sequential (NPE) and sequential version (SNPE). For evaluating SBI with simulated data, we used the NPE to leverage its ability to perform inference repeatedly for many different observations without retraining as needed for running simulation-based calibration. For the inference on the DSO rule, we used SNPE with ten rounds to focus the posterior on the measured data. For the inference on the distance-based rules, we used NPE.

**Validating SBI on the wiring rule simulator.**   We performed two validation steps to ensure that SBI performs reliably when inferring wiring rules in the structural model of the rat barrel cortex. First, we set up a simplified version of the DSO rule simulator for which it was possible to obtain a high-quality reference posterior. In particular, we reduced the number of neuron-pair-subvolume combinations from 130 million available in the original

structural model to only ten. Additionally, we omitted the summary step, such that running one simulation corresponded to applying the rule to the structural features of the ten neuron-pair-subvolume combinations and sampling synapse counts from the corresponding Poisson distribution. As a consequence, the likelihood of this simplified simulator was accessible, i.e., it was given by the Poisson distribution, and it was possible to obtain accurate posterior samples using standard approximate Bayesian inference MCMC sampling [80].

We implemented this reduced simulator in the SBI benchmarking framework `sbibm` [42] and obtained references posterior samples via slice sampling MCMC [92] using ten parallel chains and sequential importance reweighting [42, 79], all as implemented in `sbibm`. We compared three algorithms: SNPE, which estimates the posterior in multiple rounds focusing on one specific observation; the single-round variant NPE, which works for many observations without retraining; and a classical sequential rejection-sampling-based algorithm called SMC-ABC [38, 39]. As a measure of posterior accuracy, we used the classifier-2-sample-test score (C2ST, [41]), defined by the classification accuracy of an artificial-neural-network classifier trained to distinguish the approximate and reference posterior samples. For running SNPE and NPE, we used the `sbi` toolbox [33], and for SMC-ABC, we used the implementations provided in `sbibm`. Generating the training data for the SBI algorithm by simulation data from the model can be a crucial computational factor. To investigate the simulation efficiency of different SBI algorithms for our inference problem, we performed a quantitative comparison of the number of training simulations and the resulting accuracy of the approximate posterior by repeating inference with SMC-ABC, NPE, and SNPE for a simulation budget of 1,000; 10,000; 100,000 and 1,000,000 simulations.

As a second validation step, we applied SBI to the original version of the DSO rule for which no reference posterior was available. For this setting, we tested the validity of NPE applied to simulated observed data. First, we performed simulation-based calibration (SBC) to check the calibration of the posterior uncertainties inferred by NPE. To run SBC, we trained NPE once on 1,000,000 simulations and then obtained 1,000 different posteriors $p(\theta|x_i)$ for different observations $x_i$, where $x_i$ was generated from different parameters $theta_i$ sampled from the prior. We then collected the individual ranks of the underlying parameter $\theta_i$ under their posterior and tested whether these ranks were uniformly distributed by visually inspecting their empirical cumulative density functions. Second, we checked whether the parameters identified by the NPE posterior distribution could reproduce the (simulated) observed data. To perform this check, we sampled 1,000 parameters from the posterior inferred given a simulated example observation, simulated corresponding connection probabilities using the DSO rule simulator, and compared them to the observed data.

## Supporting information

**S1 Fig. Validating SBI over wiring rule parameters with simulated data.**
(TIF)

**S2 Fig. Prior and posterior predictive distributions.**
(TIF)

**S3 Fig. Conditional posterior distributions.**
(TIF)

**S4 Fig. Prior predictive distribution for the DSO rule.**
(TIF)

**S5 Fig. SBI results for the two-parameter DSO rule.**
(TIF)

**S1 Text. Prior predictive checks.**
(PDF)

**S2 Text. Alternative parametrization of the DSO rule.**
(PDF)

## Acknowledgments

We thank Auguste Schulz, Guy Moss, and Zinovia Stefanidi for their discussions and comments on a preliminary version of the manuscript.

## Author Contributions

**Conceptualization:** Jan Boelts, Hans-Christian Hege, Marcel Oberlaender, Jakob H. Macke.

**Data curation:** Jan Boelts.

**Formal analysis:** Jan Boelts, Philipp Harth, Jakob H. Macke.

**Funding acquisition:** Daniel Baum, Hans-Christian Hege, Marcel Oberlaender, Jakob H. Macke.

**Investigation:** Jan Boelts, Philipp Harth, Felipe Yáñez.

**Methodology:** Jan Boelts, Jakob H. Macke.

**Project administration:** Jan Boelts.

**Resources:** Daniel Udvary.

**Software:** Jan Boelts, Philipp Harth.

**Supervision:** Daniel Baum, Hans-Christian Hege, Marcel Oberlaender, Jakob H. Macke.

**Validation:** Jan Boelts.

**Visualization:** Jan Boelts, Philipp Harth, Richard Gao, Jakob H. Macke.

**Writing – original draft:** Jan Boelts.

**Writing – review & editing:** Jan Boelts, Philipp Harth, Richard Gao, Daniel Udvary, Felipe Yáñez, Daniel Baum, Hans-Christian Hege, Marcel Oberlaender, Jakob H. Macke.

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
