## [Decision Letter · Decision Letter 0]

2 May 2023

Dear Mr Boelts,

Thank you very much for submitting your manuscript "Simulation-based inference for efficient identification of generative models in computational connectomics" for consideration at PLOS Computational Biology.

As with all papers reviewed by the journal, your manuscript was reviewed by members of the editorial board and by several independent reviewers. In light of the reviews (below this email), we would like to invite the resubmission of a significantly-revised version that takes into account the reviewers' comments.

In general the reviewers were positive about the paper but requested more clarity on the details of the method itself, a discussion of how the approach may generalise to other related modelling domains, and for greater statistical validation and interpretation of the results.

We cannot make any decision about publication until we have seen the revised manuscript and your response to the reviewers' comments. Your revised manuscript is also likely to be sent to reviewers for further evaluation.

Sincerely,

Emma Claire Robinson

Academic Editor

PLOS Computational Biology

Mark Alber

Section Editor

PLOS Computational Biology

Reviewer's Responses to Questions

**Comments to the Authors:**

Reviewer #1: Boelts et al present the application of the simulation-based inference (SBI) framework to generative modeling of connectomes. The authors first introduce the SBI framework as applied to generative modeling, before applying it to formulate wiring rules of synaptic connectivity in rat barrel cortex. In multiple experiments, the authors evaluate SBI accuracy, study interactions between recovered model parameters and demonstrate the use of SBI to reject implausible wiring rules, using a comparison of SBI-inferred parameters to those derived from Peter’s rule.

The study is timely, thorough, and nicely illustrated. My main issue is that the methods and results are occasionally difficult to follow, and that additional interpretation of some of the results could be provided. Please find below suggestions and questions aimed at improving clarity.

Comments

Figure 1 and parts of the introduction aim to summarise the advantages of SBI over conventional approaches to generative modeling. However, it isn’t entirely clear whether the comparison is fair, for example in the comparison of a “single hypothesis about the connectome” in conventional modeling, with “multiple a-priori hypotheses” in SBI (Fig. 1). Conventional generative modeling does enable comparison of multiple competing hypotheses, through implementation of multiple candidate models. Even within a single model, multiple parameter combinations can be systematically evaluated. Similar language is used in the text, when suggesting that “conventional approaches often optimize for one best-fitting parameter”. Arguably, the conventional process can be seen as more “manual” and potentially less efficient than the SBI approach. However, the terminology used in the figure and main text seems to oversimplify the comparison.

The methodological details are dense and at times hard to follow, particularly for readers not familiar with Bayesian inference and generative modeling. In particular, the paragraph on validating SBI performance, titled “SBI performs accurately on simulated data” was difficult to follow without reference to both Materials and Methods, and Supplementary Information. I would recommend providing additional details here.

When introducing application of the model to rat barrel cortex, the authors describe that “simulating data with random parameters covers the entire range of probabilities […] including the measured data”. Perhaps I am missing something, but this seems trivial. Can the authors clarify the aim of this step? Can the authors also comment on the shape of the resulting prior distribution, heavily skewed towards the extreme values (0 and 1)?

The authors apply SBI to simulate connectomes of the rat barrel cortex, based on the “dense structural overlap” (DSO) rule, and identify parameter combinations which are compatible with the observed data. However, an interpretation of these findings seems to be missing. Can the recovered values be interpreted relative to the literature? Does the parametrized DSO rule provide additional insight over the previous form, assuming that pre- and postsynaptic features have the same relative importance? On a related note, it seems notable that the maxima of recovered parameter values in the parametrized DSO rule are highly similar to each other, also suggesting a relatively similar importance of DSO model features.

Further, the authors use held-out data to predict unobserved connection probabilities. The description and interpretation of some aspects of this experiment could be more extensive. For example, authors note that “predictions […] clustered around actual measurement values for most of the seven connection probabilities”. Statistics would be beneficial here, to quantitatively compare alignment of measures and predicted parameters. Moreover, the predictions for some neurons seem poorly aligned with measurements; a more detailed discussion of this, and of the differences in agreement between measured and predicted parameters across neuron types, would be welcome. Finally, the authors briefly mention the use of a classifier trained to distinguish between predictions inferred from all measurements, and for held-out measurements. More details should be provided here, on the aims of this, the classifier used, and interpretation of the findings (i.e. held-out predictions seemingly failing for some neuron types).

Similarly to above: during evaluation of the neuron-level and synapse-level rules, the measurement for the L4SS neuron is better aligned with Peter’s rule, than with the inferred data. It would be useful to quantify this (mis-)alignment, and certainly to discuss and interpret it.

Minor comments

The use of the term “neural networks” in the abstract might be prone to misinterpretation as relating the machine learning context of artificial neural networks, rather than the desired biological context. Perhaps a term such as “neuronal networks” or “connectomes” might be preferred?

The authors share code related to this paper on GitHub, including tutorials, which is welcome. This builds on the prior release of the “sbi” Python package, which is a laudable effort. However, the authors might wish to rephrase the last sentence of the introduction, to clarify that the “sbi” package is not being made available for the first time in this context.

Typos

“Is this specific combination of pre- and postsynaptic features in the DSO rule *is* the only valid choice?”

“…the higher *the* spatial resolution, the fewer common subvolumes were required…”

Reviewer #2: Thank you for the opportunity to review this work by Boelts and colleagues. The authors propose an interesting use of simulation-based inference (SBI) to efficiently fit parameters of connectome generative models. The paper tackles an important issue: fitting wiring rules to empirical connectivity data typically entails time-consuming sweeps of large parameter spaces. Using a combination of Bayesian inference and machine learning, the authors show—through multiple lines of evidence—that their approach can alleviate these costs by simultaneously characterising the posterior distributions of all wiring parameters.

My opinion of the work is positive. The paper was a pleasure to read—the motivation for the study is set out clearly and didactically, the figures are beautiful and informative, and the text is very well written. That said, I do have a few comments and questions below. My expertise is in network neuroscience and I have only superficial knowledge in Bayesian inference and machine learning, so most of my remarks are related to the application of the present framework to the type of generative models done in network neuroscience. I hope the authors will find these comments useful to further improve their work.

1. One of the key steps in the wiring rule simulator is "3. Calculation of summary statistics from the simulated connectome to match the available measurements". Here, the authors analysed a cellular-level network of the rat barrel cortex, and the fit to empirical data was measured as the ability to recapitulate observed connection probabilities between different cell types. Instead, in the network neuroscience literature (e.g., Betzel 2016, Vertes 2012; all properly cited), the typical goal is to match complex topological properties of empirical connectivity. That is, to generate synthetic networks with, e.g., degree, connection-centrality, or connection-length distributions that match those observed in empirical brain networks. In many cases, researchers seek to match multiple properties at once, and do so by defining multi-objective energy functions. Can the SBI framework be applied in these—arguably—more complex scenarios? The computation of these network measures can be time consuming, and I imagine it could pose certain barriers to the estimation of parameter posterior distributions? To be clear, I think the present focus on cell type connection probabilities is perfectly fine, I am just trying to gauge the applicability of this framework to different lines of connectome research. I would appreciate it if the authors could comment on this point.

2. Is there a distinction to be made here between estimating all parameter values for a single wiring rule vs. finding all potential wiring rules? In several places in the manuscript, including the author summary (lines 45–47), it is implied that these two things are the same. From my understanding, SBI allows for an efficient exploration of all parameter values for a pre-specified wiring rule. I do not think the authors provided any evidence that their framework allows for a simultaneous exploration of all possible wiring rules—indeed, that would be very hard, as there are seemingly infinite ways to formulate a wiring probability equation. To come back to the network neuroscience literature, a lot of work has been done is comparing different wiring rules, i.e., connectivity growth equations constructed based on different topological properties of the networks. I currently don't see how SBI could handle the automation of this comparison. Could the authors comment on this?

3. Another important point is that the current generative model does not seem to be an iterative growth model. In many network-based generative models, connections are added one at a time. This is important because as the network topology evolves, the addition of each new connection influences the probability of wiring between nodes. In many lines of research this is an important feature of connectome generative models, and it provides an opportunity to study questions relating to biological development over time (see DOI:10.1126/sciadv.abm6127, for example).

4. Might be worth mentioning that virtually all previous work on human connectome generative models also provide evidence against Peters' rule—that is, adding topological information into the wiring rule almost always improves on a naive model based solely on inter-regional distances.

5. A recent reference that should be discussed: https://www.sciencedirect.com/science/article/pii/S1053811923001088

**Have the authors made all data and (if applicable) computational code underlying the findings in their manuscript fully available?**

Reviewer #1: Yes

Reviewer #2: None

PLOS authors have the option to publish the peer review history of their article (what does this mean?). If published, this will include your full peer review and any attached files.

Reviewer #1: No

Reviewer #2: No
---

## [Decision Letter · Decision Letter 1]

1 Aug 2023

Dear Mr Boelts,

We are pleased to inform you that your manuscript 'Simulation-based inference for efficient identification of generative models in computational connectomics' has been provisionally accepted for publication in PLOS Computational Biology.

Best regards,

Emma Claire Robinson

Academic Editor

PLOS Computational Biology

Mark Alber

Section Editor

PLOS Computational Biology

Reviewer's Responses to Questions

**Comments to the Authors:**

Reviewer #1: The authors have addressed all of my comments, and did a excellent job making the paper clearer. Great work!

**Have the authors made all data and (if applicable) computational code underlying the findings in their manuscript fully available?**

Reviewer #1: Yes

PLOS authors have the option to publish the peer review history of their article (what does this mean?). If published, this will include your full peer review and any attached files.

Reviewer #1: **Yes: **Frantisek Vasa

---

## [Editor Report · Acceptance letter]

14 Sep 2023

PCOMPBIOL-D-23-00355R1 

Simulation-based inference for efficient identification of generative models in computational connectomics

Dear Dr Boelts,

I am pleased to inform you that your manuscript has been formally accepted for publication in PLOS Computational Biology. Your manuscript is now with our production department and you will be notified of the publication date in due course.

With kind regards,

Zsofi Zombor
